# Understanding the role of household hygiene practices and foodborne disease risks in child stunting: a UKRI GCRF Action Against Stunting Hub protocol paper

Paula Dominguez-Salas [1,2] Hugh Sharma Waddington,[3] Delia Grace,[1,4] Caroline Bosire,[4] Arshnee Moodley,[5] Bharati Kulkarni,[6] Teena Dasi,[6] Santosh Kumar Banjara [7] Ramachandrappa Naveen Kumar [6] Umi Fahmida [8] Min Kyaw Htet [8] Arienta R P Sudibya,[8] Babacar Faye,[9] Roger C Tine,[10] Claire Heffernan,[11,12] Deepak Saxena,[13] Robert Dreibelbis,[3] B Häsler[14]

**Correspondence to**
Dr Paula Dominguez-Salas; P.
DominguezSalas@gre.ac.uk

## ABSTRACT

**Introduction** Environmental hygiene and food safety are important determinants of child stunting. This research aims to explore the relationship between child stunting and household hygiene practices and behaviours, including the availability of water, sanitation and hygiene (WASH) facilities; the use of safe food and good quality drinking water (especially when used for complementary feeding); hygienic practices in food transport, storage and preparation and the control of cross-contamination from animals, their produce and waste.

**Methods and analysis** This study is part of a wider observational study which aims to investigate the interdisciplinary factors contributing to child stunting using a 'whole child' paradigm. The observational study recruits women during pregnancy in Hyderabad, India, Lombok, Indonesia and Kaffrine, Senegal, and dyads (ie, 500 mother–infant pairs per country) are followed longitudinally up to 24 months after birth. Within the interdisciplinary niche, the study here has developed tools to investigate the potential exposure pathways to environmental pathogen contamination of foods and water. Holistic WASH and food safety data collection tools have been developed to explore exposure pathways at the household level, including: (1) survey questionnaires; (2) spot-checks; (3) biological sampling of drinking water, food and domestic surfaces and (4) direct observation. An integrated analytical approach will be used to triangulate the evidence in order to examine the relationships between child stunting, WASH and food safety behaviours.

**Ethics and dissemination** Ethical approval of the study was granted by the ethics committee of the LSHTM, RVC, ILRI, ICMR, IIPHG, SEAMEO-RECFON, University of Cheikh Anta Diop. Findings of the study will be disseminated through publication in peer-reviewed journals, relevant international conferences, public engagement events, and policy-maker and stakeholder events.

## WHAT IS ALREADY KNOWN ON THIS TOPIC

⇒ From the moment complementary feeding begins, the chances of exposure to food contamination and infectious disease increase exponentially.
⇒ Adequate water, sanitation and hygiene (WASH) and food safety practices can play a key role in reducing infectious disease transmission and child malnutrition.
⇒ The contribution of hygiene behaviours and food contamination in driving transmission of infections and childhood stunting requires further investigation.

## WHAT THIS STUDY ADDS

⇒ Development, application and analysis of integrated WASH and food safety tools, towards a holistic understanding of how these practices and behaviours impact child stunting.
⇒ Evidence towards a new typology of child stunting, with a focus on exposure pathways for pathogens, and to support programmes and policies to minimise stunting in childhood.

## HOW THIS STUDY MIGHT AFFECT RESEARCH, PRACTICE OR POLICY

⇒ Understanding of the role of home hygiene and food safety related practices in childhood stunting with a view to identify solutions.
⇒ New and more integrated approaches in research and practice that bring together WASH and food safety considerations at the household level to investigate exposure to pathogens.
⇒ Evidence on complementarities between food hygiene and WASH in order to help design suitable interventions to combat stunting.

## INTRODUCTION

The incidence and severity of stunting in childhood is closely linked to exposure to

infection from food, drinking water and the wider environment.[1] Infections associated with contaminated water or insufficient water, sanitation and hygiene (WASH) are responsible for an estimated 21% of the total global burden of diseases,[2] contributing to the outbreak and chronicity of preventable infections such as diarrhoeal diseases and acute respiratory infections, which are the two leading causes of death in children globally.[3] However, three recent randomised controlled trials have assessed the impacts of WASH combination interventions on nutrition in Bangladesh,[4] Kenya[5] and Zimbabwe.[6] The studies were not able to detect any effects on child linear growth, and only in Bangladesh was diarrhoea reduced.[7] Cumming *et al* have challenged the efficacy of the WASH interventions in addressing common causes of faeco-oral disease in low-income contexts, such as cryptosporidium.[8] More generally, the interventions did not increase the quantity of water supply available to practice domestic hygiene.

In addition, the WHO Foodborne Disease Burden Epidemiology Reference Group (FERG), estimated that approximately 40% of the global burden of foodborne diseases occurs in children under the age of 5 years.[9] A common cause of foodborne infections is the consumption of raw or undercooked meat, fish, seafood, eggs, fresh produce and dairy products contaminated by norovirus, Campylobacter, non-typhoidal Salmonella or pathogenic *Escherichia coli*.[9] Food hygiene is a primary concern in the preparation of complementary foods—for example, bottles often cannot be adequately sterilised and perishable complementary foods are often left unrefrigerated or eaten using unclean utensils or with unwashed hands.

Disease transmission from faeces can be foodborne or waterborne (ie, oral ingestion of contaminated food and water), water washed or water scarce (ie, spread through inadequate hand and food hygiene) and water based (ie, transmitted by parasites that penetrate skin in water, such as schistosomiasis or by walking barefoot on contaminated soil in the case of hookworm).[10] For example, theory and evidence suggests that pathogens in foods are a main driver of infant faecal-oral disease.[11] Field studies have demonstrated high bacterial contamination from *E. coli* or Salmonella in weaning foods.[12 13] Of particular importance for the cleanliness of infant complementary foods are the quality of water, surfaces and utensils used to prepare and eat them[14–16] as well as the quality of the food ingredients procured.

Hence, the availability of WASH and food storage and preparation facilities is likely to determine exposure to infection. Nonetheless, pathogen exposure can also arise when available facilities are used inadequately. For example, cultural norms and beliefs can restrict domestic food and hygiene practices—for example, whether mothers are the principal decision-makers about food preparation and hygiene in the household, or whether women are able to use the same toilet facilities as other household members.[17] However, while contamination of foods during cooking and feeding is common, the behavioural drivers have not been adequately explored. A recent systematic survey of 350 WASH evaluation projects in low-income and middle-income countries found that around 15% of studies examined nutrition outcomes like stunting, and handwashing before food preparation, but only 5% reported on other food hygiene behaviours, such as whether food was stored appropriately and kitchen utensils were washed.[18]

While food safety and WASH are interlinked, to date little attention has been paid to understanding the relationships and trade-offs between food safety, WASH and child stunting. The safety of livestock and fish derived foods is a particularly neglected area and there has been no systematic integration of these topics in food safety and WASH protocols.[14] The UKRI GCRF Action Against Stunting Hub (AASH) explores child stunting from the Whole Child Approach, that is, considering different aspects of child development, including biology considerations, the home environment, the educational environment and the food system, that can contribute to thriving childhood. The aim of this study is to examine WASH and food safety elements, their convergence and contributions to child stunting at the household level. It evaluates WASH and food safety practices across households in three study sites with the objective to provide evidence on potential exposure to pathogens through contaminated environment, drinking water, and complementary food, and associated risk factors.

## METHODS AND ANALYSIS
### Conceptual framework

The manner in which WASH and food safety at the household level can potentially lead to changes in infection and dietary intake, thereby affecting child stunting, is illustrated in figure 1, adapted from the UNICEF's (1990) conceptual framework on the causes of malnutrition.[19] Inadequate WASH practices and food safety can contribute to childhood stunting directly, by exposing infants to infection via complementary foods contaminated during preparation and storage, or through feeding with unclean utensils, including the hands of carers. The safety of foods can also be compromised before arrival at households, throughout the different steps and processes of the food value chain. Those food hazards can be intrinsic to the food produced (eg, pathogens from animal diseases, heavy metal marine contamination), or extrinsic through processing and handling (eg, inadequate hygienic practices, aflatoxin contamination during storage).[20] Mechanisms include nutrient loss (eg, during diarrhoeal episodes), energy used by the immune system (eg, in combating illnesses), and increased intestinal permeability and reduction in nutrients absorption (eg, enteropathy).[21]

In theory, access to adequate WASH facilities enables appropriate drinking water, hygiene and food safety practices, contributing to reduction in transmission of

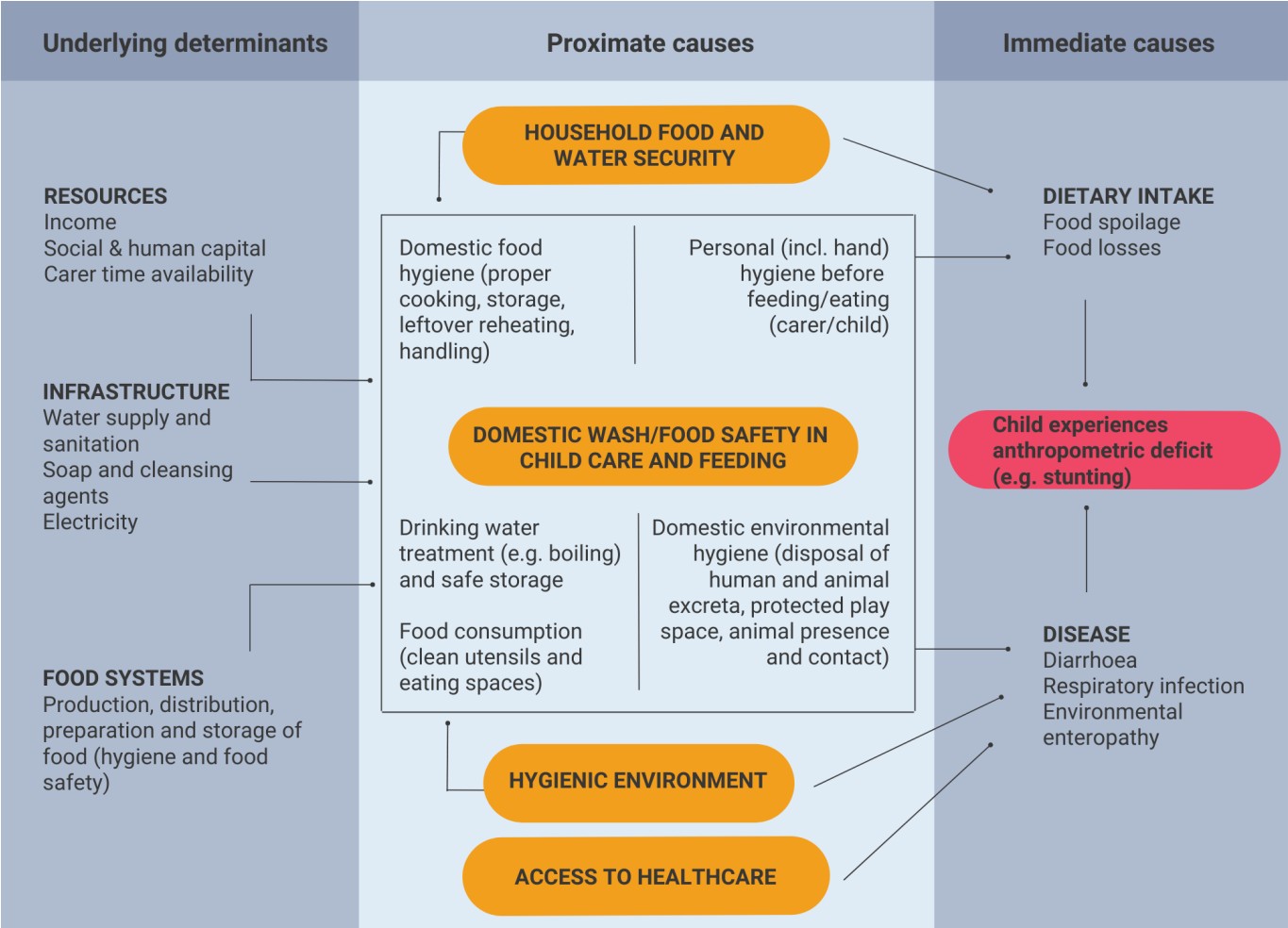

**Figure 1** A conceptual framework showing the linkages of WASH and food safety with stunting at the household level. WASH, water, sanitation and hygiene.

foodborne and waterborne pathogens and improving nutrient uptake, and ultimately child growth and development.[20] Drinking water may be contaminated at the point of supply, or through transport and storage.[15 16] The availability of clean water is critical for optimal hygiene practices,[10] to address contamination by human and animal faeces,[22] and to prepare food hygienically. Sanitation facilities can enable safe faeces disposal, especially those of young children, which can be particularly pathogenic.[23] Faecal contamination can also come from the presence of domestic and wild animals or pests inside the household or in surrounding spaces.[24] It is expected that these sources of infection become more important as a child starts crawling and interacting with their environment. As such, understanding of the determinants of food and hygiene practises in the household is key to improving infant health and nutrition.[25]

Along with direct exposure, there is also an indirect route from water supply and sanitation facilities to childhood malnutrition, which operates through the opportunity costs of resource use. The time carers have to observe, correct, and address unhygienic behaviour, and the scarcity of time in relation to competing activities that need to be done by carers in the home or workplace, may also be detrimental for children's nutritional outcomes.[26] For example, it is typically women and older children, both of whom are more likely than others to be carers of young children, who spend time fetching water, engaging with animal fodder and collecting firewood. This may occur at the expense of adequate childcare and feeding, food preparation, education or rest.[27] When drinking water needs cannot be met affordably, households may need to purchase less food in order to afford enough water to drink. This kind of trade-off is clearly more prevalent for the poorest households, for whom malnutrition is also likely to be highest.

The WASH and food safety approaches, including their scope, evolution and methods are described in table 1. Combining both WASH and food safety approaches provides an opportunity for more effective integration and learning of the two disciplines and their contributions to child growth and development. The two should converge when looking at household units with a central focus such as child stunting.

**Table 1** Characteristics of water, sanitation and hygiene (WASH) and food safety and veterinary public health research

| | WASH | Food safety and veterinary public health |
|---|---|---|
| Scope | To investigate deficiencies in water supply and quality, sanitation and hygiene to improve public health. Also, to evaluate the adequacy of hygiene standard to minimise foodborne diseases risks, and the role of water supply and sanitation access as enabling factors for improving child care, health and nutrition. | To pursue a comprehensive 'farm to fork', 'stable to table', 'boat to throat' approach to prevent and minimise risks of food contamination at all stages of the food chain from production to human consumption and waste management. |
| Sector | Human-centred with a strong anchor in the public health sector. | Focus on animal source foods, zoonoses and their respective risks. |
| Evolution | Historically, WASH research has focused on water-related disease transmission from human excreta (faecal-oral diseases). This has contributed to understanding the mechanisms of faecal-oral infection through waterborne, water-washed and water-based routes. Early intervention studies focused on the provision of infrastructure, and information, education and communication. Current research focuses more on behaviour change communication using bottom-up approaches, and incorporating the 'A (Animals)' into WASH more holistically.[24] | Veterinary public health traditionally looked at all food chain stages to identify where risks (eg, biological, chemical and physical) can emerge and how to prevent or mitigate them. The initial focus was on microbiological aspects with a strong technical dimension. Veterinary public health now looks at human behaviour more widely, bringing in social science aspects and economics to understand better practices and the motivations behind them. There is a (slow) move towards systems thinking, and a lot of progress in quantitative microbiological and epidemiological studies, detection techniques, etc. |
| Research methods | ► Interventions (eg, randomised controlled trials). <br> ► Cross-sectional, questionnaire- based surveys. <br> ► Longitudinal questionnaire-based surveys. <br> ► Spot checks. <br> ► Direct observation. <br> ► Biological sampling and testing. <br> ► Risk assessments. <br> ► Qualitative research on behaviour, practices and perceptions, including participatory approaches. | |

## Study design

This is a multicountry longitudinal observational study where women are recruited during pregnancy in Senegal, Indonesia and India and mother–infant pairs are followed up to 24 months after birth. Average stunting prevalences in these three countries selected are medium (17.9% in Senegal) to very high (30.8% in Indonesia and 37.4% in India).[28] This protocol involves multiple approaches to assess WASH and food safety practices in urban or rural households of women enrolled in the study, in Hyderabad city, India, East Lombok, Indonesia and Kaffrine, Senegal, at predefined time points (see AASH observational cohort description in Heffernan *et al* in this supplement, including further details on the study setting, sampling process, participant schedule, data collection and quality control, data management, etc).

## Data collection

This WASH and food safety data collection protocol focuses on three areas: exposure pathways to wider environmental pathogens, foodborne hazards and waterborne hazards. Four data collection tools for the household-level have been developed, namely: (1) a module for WASH and food safety to be included to the cohort questionnaire-based interviews (see online supplemental appendix 1); (2) spot-checks (see online supplemental appendix 2); (3) biological sampling and (4) direct observation (see

online supplemental appendix 3). The interviews and spot-checks take place simultaneously during home visits at 9, 12 and 18 months after birth. Data will be collected throughout the different seasons. Direct observation and biological sampling are conducted during a separate subsequent visit at each time point, in a subsample of the cohort. No additional inclusion or exclusion criteria were considered in the selection. Food and water samples are kept to a minimum for the analysis required, namely half filling a 50 mL Falcon tube with food served to the child, and the measure of the container in which the child is normally given drinking water, respectively.

The food safety component of the data collection focuses on nutrient-rich foods with the potential to alleviate stunting, particularly on animal source foods (ASFs). This is due to their important nutritional profile,[29] their documented potential to alleviate stunting[30–33] and their high food safety risk profile. Selected ASFs will be considered at each country, according to their relevance.

Trained enumerators collect data using electronic tablets in all study households, as summarised in table 2. The direct observation tool is recorded on paper and enumerator positioning and previous rapport creation are used as ways to minimise bias. Privacy is maintained by stopping the direct observation on the respondent's request. Data collection tools developed in English are

**Table 2** Summary of the integrated study methods and tools used to assess WASH and food safety at the household level

| Tool | Description | Sample size |
|---|---|---|
| Questionnaire-based interview | Mothers are asked about infant and young child feeding practices, household food expenditure and the main household decision-makers and influencers of these activities. Physical activity questionnaires ask mothers about the times spent on work and housekeeping including collecting water, fodder and fuel. Furthermore, there are questions on water sources, water security, sanitation and handwashing. Other questions focus on the practices of acquisition, transport, storage, preparation of ASF and the points of potential contamination in the household. Finally, other questions refer to the presence and behaviour of animals and potential transmission pathways associated with close proximity to animals, such as direct contact between animals and children or food. The hazards and risks prior to arrival of ASFs in the household are elucidated in a separate component, at the value chain level (see Cooper *et al* in this supplement). | Whole cohort (500 households per country) |
| Spot-check | Enumerators observe behavioural issues alongside the interview questionnaire. The data are recorded using a check list on the availability and type of sanitation facilities available at the households (eg, the place for defecation and proximity), handwashing infrastructure (eg, location, availability of water and soap), presence of animals (eg, livestock, pests), food storage facilities and practices (eg, functioning refrigerator) and food preparation (eg, easy to clean equipment). | Whole cohort (500 households per country) |
| Direct observation | Enumerators stay at the household for 3 hours around a feeding event, observing the mothers perform their daily activities and taking a passive approach to blend into the background in order to better observe hygiene related behaviours. Structured observation include activities related to water collection and storage, as well as personal, environmental and food hygiene (food preparation, storage, child feeding, handwashing and animal contact). This data collection method is a gold standard for actual observation of behavioural practices. | Subsample (approximately one-third of full cohort in each country) |
| Biological sampling and testing | Enumerators collect samples of stored water, foods to be consumed by the children (as close to the feeding of the child as possible) and a swab of the main food preparation area at the households. Samples of hands rinse are optional. The samples are stored in a coolbox and transported to a specialised laboratory for analysis. The samples are processed in-country and tested for overall contamination and for selected pathogens (such as Salmonella, Shigella, *Escherichia coli* and Campylobacter for food and *E. coli*/faecal indicators for surfaces and water), using standard laboratory protocols (of conventional culture and PCR). Part of the samples collected are used for lab analysis and the remaining part stored under –20°C for future analysis and crosschecking. | Subsample (approximately one-third of full cohort in each country) |

ASF, animal source foods; WASH, water, sanitation and hygiene.

translated into local languages and backtranslated into English to check for accuracy, and piloted in the field.

### Data management and analysis

Data are collected at the households in CommCare and checked for accuracy before being transmitted to a central repository and can be downloaded in Excel for analysis. Samples are analysed at laboratories in each of the countries using tailored standard operating procedures.

Descriptive, comparative and regression analyses will be used to investigate the impact of WASH and food safety on child stunting. Factors relating to WASH and food safety amenities and practices will be considered simultaneously as external risks or preventive factors influencing the incidence and severity of stunting, using multivariable statistical approaches. To relate the main exposure variables (WASH and food safety factors) with the main outcome (stunting, measured as either z-scores or presence/absence), multivariable regressions models will be developed. According to our causal framework presented in figure 1, we will adjust this model for potential confounding factors. Some of these confounding factors will be fairly stable (eg, sex, maternal literacy, household facilities, socioeconomic status) and others will be time-varying (eg, age, breastfeeding, environmental hygiene levels). We will avoid intermediate factors (eg, infection rates). Subsequently, we will explore the potential role of the mediators in the causal framework using mediation analysis. Due to the existence of repeated measures conducted over three different time points for each child, we will account for this with a random effect term by individual in the regression model. The model will be a linear regression for stunting measured as a continuous variable, and a logistic (or ordered logistic) regression for stunting (moderate and severe) measured as a dichotomous variable. Due to the fact that in our three

country settings the relevant association might differ, we will estimate the models separately in each country, and afterwards we will conduct individual participant data meta-analysis of the coefficients of interest across the countries. This strategy will allow for an optimal fit of the model in each country and at the same time we will be able to explore the heterogeneity of coefficients across countries and within subgroups (by age, sex and socio-economic characteristics).

Data triangulation will also be used, first, to address biases in individual data collection tools. For example, known biases occur when measures are reported by participants, particularly by those people for whom illness becomes normalised after repeated bouts of infection.[34] Observation of behaviours, through enumerator presence in the household at certain times of day, can help to mitigate reporting bias. However, surveys and observation may also affect participant motivation, particularly when repeated over long periods. Extensive collection of samples for laboratory testing can be done, but have their own weaknesses.[35] Second, data triangulation through statistical modelling will be used to determine the relationship between WASH and food safety in each of the study contexts, and to explore relevant questions such as whether they are complements or substitutes in child health and nutrition. As such, a mix of approaches is being used in this study to address these potential challenges.

### Patient and public involvement statement

Study participants have not been involved in the general design of the study. However, the study was designed in partnership with local research partners and stakeholders with experience in working at each of the settings. Local enumerators helped finalise the data collection tools during training and pilot testing.

Extensive consultations will be organised with the local communities to understand the ethos of the research including the involvement of participants and the implications of the research. Question and answer sessions will be held during the consultations to answer any questions and/or concerns raised. Study participants and the public will be involved in the dissemination of the study's findings through community discussion or engagement events.

### DISCUSSION

Poor food safety and inadequate WASH, including getting in direct contact with animals, their produce and/or their faeces, are considered as important contributors to undernutrition. Cumulatively over the first 2 years of life, these factors can determine linear growth faltering. However, the causes of infant stunting are not exclusive to these factors, and therefore, it is critical to cohesively consider the myriad of determinants involved. The UKRI GCRF AASH is using an interdisciplinary approach to investigate this variety of factors causing stunting with the aim to identify solutions for the amelioration of growth and development in early childhood at the 'whole-child' level. Within such an interdisciplinary framework, this study design will generate comprehensive evidence on the contribution of WASH, and food safety at the household level.

Challenges to conducting integrated assessment in WASH and food safety in a multicountry study like this are various, including identifying key ASFs and key hazards in a systematic way, or the analysis of pathways linking hygiene, food and water safety, and sanitation data into the typologies of stunting. For this purpose, we developed holistic data collection tools relevant to each study setting, to measure risk factors for exposure to foodborne and waterborne disease. Multiple household-level interventions are available in both WASH and food safety but, to date, their effectiveness has not been assessed together, despite the similarity in the exposure mechanisms they are targeting. The effectiveness of relevant interventions and technologies in addressing malnutrition depends on both their biological efficacy, and the degree to which consumers adapt to and use the new approach to their contexts.[8] We propose an approach and suggest data collection tools through which these various underlying and proximate determinants of child growth may be analysed.

The data generated from the study will help pave the way for future interventions to manage factors associated with and aimed at preventing stunting. At the individual level (immediate causes), inadequate practices and knowledge gaps identified will help tailor education and social behaviour change communication interventions. These are aimed at improving intake while reducing disease risks. At the household level (proximate causes), the findings can inform the implementation of more efficient WASH interventions, that ensure healthy environments and better personal hygiene. Ultimately, it can inform advocacy efforts to improve infrastructures and services at the community, regional or even national level (underlying causes).

ASF value chains generally present important potential to improve availability, accessibility, affordability and safety for the most nutritionally vulnerable population groups.[36] This study can provide information on mechanisms through which value chains are linked to stunting. The study also offers information that will help to identify entry points in chains where upgrading may be feasible and agreeable among stakeholders (eg, consumers, government and businesses). Based on secondary dietary and nutritional data, a systematic literature review of food safety hazards in ASF value chains in each study setting and local experts' feedback, commodities of likely relevance are eggs in all countries, milk in India and Senegal, and fish in Indonesia (see AASH food system protocol description in Cooper *et al* in this supplement for value chain work).

All in all, the described data, combined with data from other workstreams related to epigenetics, microbiome,

gut health and nutrition, education, etc also described in this supplement, this work aims to providing with a powerful piece of evidence for improving the child environment with a 'Whole child' approach.

## DISSEMINATION

Findings of the study will be disseminated through publication in peer-reviewed journals, presented at relevant international conferences, public engagement events, and policy-maker and stakeholder events. Data generated from the study will be deposited in a publicly available data repository.

**Author affiliations**
[1]National Resources Institute, University of Greenwich, London, UK
[2]Policies, Institutions and Livelihoods Programme, International Livestock Research Institute (ILRI), Nairobi, Kenya, Nairobi, Kenya
[3]Department of Disease Control, London School of Hygiene and Tropical Medicine, London, UK
[4]Animal and Human Health Programme, International Livestock Research Institute (ILRI), Nairobi, Kenya
[5]CGIAR Antimicrobial Resistance Hub, International Livestock Research Institute (ILRI), Nairobi, Kenya
[6]National Institute of Nutrition, Indian Council of Medical Research, Hyderabad, Telangana, India
[7]Clinical Division, National Institute of Nutrition, Hyderabad, Telangana, India
[8]Southeast Asian Ministry of Education Organisation Regional Centre for Food and Nutrition (SEAMEO RECFON), East Jakarta, Indonesia
[9]Department of Parasitology, Université Cheikh Anta Diop (UCAD), Dakar, Senegal
[10]Department of Parasitology-Mycology, University of Cheikh Anta DIOP, Dakar, Senegal
[11]Department of Pathobiology and Population Sciences, University of London, London, UK
[12]London International Development Centre, London, UK
[13]Public Health Foundation, Indian Institute of Public Health Gandhinagar (IIPHG), New Delhi, Delhi, India
[14]Department of Pathobiology and Population Sciences, Royal Veterinary College (RVC), Hatfield, UK

**Contributors** All authors were involved in the study design and the preparation of the study tools and protocols at different stages; BH, HSW and PD-S drafted the manuscript sections; all other authors critically reviewed the manuscript. PD-S and HSW share first authorship. We want to thank Alessia Gasco for producing the conceptual framework, David Prieto-Merino for his advice on statistics, and Kaitlin Conway and Modou Jobarteh for careful editing of the manuscript.

**Funding** This work was supported by the UKRI-GCRF Action Against Stunting Hub (Project ref. MR/S01313X/1).

**Disclaimer** The views expressed in this manuscript are those of the authors and do not necessarily represent the views of the funders.

**Competing interests** None declared.

**Patient and public involvement** Patients and/or the public were involved in the design, or conduct, or reporting, or dissemination plans of this research. Refer to the Methods section for further details.

**Patient consent for publication** Not applicable.

**Ethics approval** Ethical approval for the study was granted by the Institutional ethics committees of the London School of Hygiene and Tropical Medicine (17915/RR/17513), the Social Science Research Ethical Review Board at the Royal Veterinary College (URN SR2020-0198) and the International Livestock Research Institute Institutional Research Ethics Committee (ILRI-IREC2020-33). In-country approvals were also granted: the National Institute of Nutrition (ICMR), Ministry of Health and Family Welfare, Government of India (CR/04/I/2021), the Health Research Ethics Committee, University of Indonesia and Dr Cipto Mangunkusumo Hospital (KET-887/UN2.F1/ETIK/PPM.00.02/2019), and the Comité National d'Ethique pour la Recherche en Santé, Senegal (Protocole SEN19/78). Mothers and relevant family members are explained to all the procedures and processes prior to acceptance, then informed consent is obtained. All data and samples are anonymised with a unique identification number. Withdrawal from the study is allowed at any stage.

**Provenance and peer review** Not commissioned; externally peer reviewed.

**Data availability statement** No data are available.

**ORCID iDs**
Paula Dominguez-Salas http://orcid.org/0000-0001-8753-4221
Santosh Kumar Banjara http://orcid.org/0000-0002-0893-9552
Ramachandrappa Naveen Kumar http://orcid.org/0000-0002-8984-5741
Umi Fahmida http://orcid.org/0000-0003-1403-6242
Min Kyaw Htet http://orcid.org/0000-0001-6417-2942

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
