## [Reviewer comments · BMJ Paediatrics Open]

ARTICLE DETAILS

TITLE (PROVISIONAL)	Understanding the role of household hygiene practices and foodborne disease risks in child stunting: a UKRI GCRF Action Against Stunting Hub protocol paper
AUTHORS	Dominguez-Salas, Paula Waddington, Hugh Sharma Grace, Delia Bosire, Caroline Moodley, Arshnee Kulkarni, Bharati Dasi, Teena B, Santosh Kumar Kumar, Ramachandrappa Naveen Fahmida, Umi Htet, Min Kyaw Sudibya, Arienta R.P. Faye, Babacar Tine, Roger Heffernan, Claire Saxena, Deepak Dreibelbis, Robert Häsler, Barbara N

VERSION 1 - REVIEW

N/A

VERSION 2 – REVIEW

REVIEWER	Reviewer Name: Ms. Claire Grisaffi Institution and Country: Cranfield University Cranfield School of Water Energy and Environment, United Kingdom of Great Britain and Northern Ireland
REVIEW RETURNED	28-Nov-2022

GENERAL COMMENTS	Many thanks for the opportunity to review this protocol. It outlines an important and novel cross sector study, which addresses a critical research gap and responds to growing calls for more integrated understanding. The methodology brings together complimentary data collection methods to deepen knowledge and reduce bias. It would be helpful to clarify the scope of the protocol and which elements are already established under the wider observational study referenced. A number of areas for development are given below, guided by the SPIRIT checklist to be in line with BMJ requirements. If the authors believe that this is not an appropriate measure, and/or if many of the elements requested are already
---

given under existing protocols, please do reference the framework and protocols used.

It would be worthwhile to further develop the document in the following main areas, aiming for sufficient detail to allow replication:

(1) Briefly describe the study setting, sampling process, inclusion or exclusion criteria for participants.

(2) Include plans for data collection and management, including any processes to promote data quality, for example:

(a) the process for translation, back translation and piloting tools in each location

(b) an outline of the requirements for selection and training of data collectors

(c) how data collection takes seasonal or other factors into account

(d) processes for field supervision, validating and cleaning data

(3) Describe the limits on reliability for each of the tools to be used, and more detail on how they compliment each other to increase overall validity

(4) Provide an overall time schedule for participants, shown as a schematic diagram and a justification for the intervals planned

(5) Further detail on the process for managing retention - and if necessary replacement - of participants

(6) Outline of anticipated sources of bias, not avoided or mitigated by the planned actions above, and how they will be monitored or addressed

(7) A review of safeguarding issues and how they are managed for both data collectors and study participants

The only other significant comment for review is that the conceptual framework does not reference the major trials published in The Lancet Global Health, which have reported no effect of WASH interventions on linear child growth. It would be useful to recognise the debate and also describe how this study is designed to avoid the challenges faced in previous trials. A recent review is given here: Pickering, A. J., Null, C., Winch, P. J., Mangwadu, G., Arnold, B. F., Prendergast, A. J., ... & Humphrey, J. H. (2019). The WASH Benefits and SHINE trials: interpretation of WASH intervention effects on linear growth and diarrhoea. *The Lancet Global Health*, 7(8), e1139-e1146.

Some further detailed points are given below for each section:

Introduction: The 'Whole Child Approach' referenced should be briefly defined.

Conceptual framework: Consider additional justification and referencing for the more novel parts of this framework. For example, Page 9 line 4: the evolution of WASH research to incorporate animals as the 'A' in WASH.

Methodology: It could be useful to map the different sections of the tools against the study outcomes, to demonstrate that there are no gaps or duplication.

Discussion: The point raised around entry points in chains for upgrading ASF (Page 13, lines 25-28) is just one of the many areas where this study would contribute. It would be helpful to review the structure of this section so that it more clearly outlines the expected impact of findings for each different actor, linking back to the conceptual framework. This restructure and expansion could help to

	avoid the early repetition and more clearly demonstrate the importance of the study. Ethics and dissemination: Linked to the comment on safeguarding above, it would be helpful to expand this section to briefly cover the major ethical issues you expect to face and main mitigating actions to be taken. Appendices: It would be useful to include a brief preamble and guidance for each tool. Page 46: Data log - water samples: consider including checks on the water source and condition, physical parameters of the sample etc.
--	--

REVIEWER	Reviewer Name: Dr. Emmanouil Bagkeris Institution and Country: University College London, Great Ormond Street Institute of Child Health,
REVIEW RETURNED	30-Nov-2022

GENERAL COMMENTS	1. It is important to be explicit whether multilevel linear regression models will be used to account for the repeated measures over time (if needed). Will the outcome (child stunting) be assessed at a single time point (18 months)? Please clarify. This needs to be crystal-clear as it will determine the type of analysis that needs to be performed. All time-varying factors that will be assessed as confounders in the regression models need to be listed in the methods section. 2. It was not clear in the methods section whether a sample size estimation has been performed to secure that the study will have enough power to assess the association of household hygiene practices and foodborne disease risks with child stunting. If this has not been done the authors need to explain why and clarify what the prevalence of child stunting is each regions site.
---

VERSION 2 – AUTHOR RESPONSE

Re: Understanding the role of household hygiene practices and foodborne disease risks in child stunting: a UKRI GCRF Action Against Stunting Hub protocol paper

On behalf of the author team of the forenamed manuscript, I would like to take this opportunity to thank both reviewers for their feedback on our paper for this special supplement on the UKRI GCRF Action Against Stunting Hub (AASH). We have incorporated the different comments indicated by them, and discussed the changes or additional aspects, below. We have indicated the modifications made in track-changes.

Overall, we agree with the reviewers' comments and we feel that our paper has greatly benefited from the review process. We detail the steps we took to address the comments and the changes made below. Reviewers' comments are highlighted in bold italic and our responses are in grey (including relevant section or page numbers in the revised 'tracked-changes' manuscript). Please also note that the word count of this revised manuscript now exceeds slightly the initial submission limit, owing to the corrections made in response to the reviewers' comments, but it has been kept to a minimum.

We hope the changes to the manuscript and the detailed explanations will be agreeable to both, BMJ editors and reviewers. We hope that the revised protocol will be satisfactory for publication on BMJ Paediatrics Open, but please do not hesitate to get back in touch with us if you require any clarification or require any further change.

Also, regarding the additional set of comments from the editorial office, we have added the updated key messages missing on how this study might affect research, practice or policy, and we have adjusted the uploading of the manuscript as required (i.e. file designation, grant award designation, figures format, table length, and appendix format). Also, regarding the quality of the Figure in pdf, following further advice from BMJ editorial team, the quality was deemed to be ok as per email on 16th February, so it has not been modified any further.

Regarding the addition on the authorship, please, note that there has been an addition on the authorship, namely the Principal investigator of the project, Claire Heffernan. The attached complete form was circulated to all authors, and all approvals have been received.

We thank you again for your interest in our protocol, and we take the opportunity to wish you a happy 2023!

VERSION 3 – REVIEW

REVIEWER	Reviewer Name: Ms. Claire Grisaffi Institution and Country: Cranfield University Cranfield School of Water Energy and Environment, United Kingdom of Great Britain and Northern Ireland
REVIEW RETURNED	13-Mar-2023

GENERAL COMMENTS	Thank you for the comprehensive feedback and updated manuscript. No further comments.
---

REVIEWER	Reviewer Name: Dr. Emmanouil Bagkeris Institution and Country: University College London, United Kingdom of Great Britain and Northern Ireland
REVIEW RETURNED	15-Mar-2023

GENERAL COMMENTS	No further comments. The authors have adequately addressed my comments.
---

VERSION 3 – AUTHOR RESPONSE

N/A